# Nitrogenous Fertilizer Levels Affect the Physicochemical Properties of Sorghum Starch

**DOI:** 10.3390/foods11223690

**Published:** 2022-11-17

**Authors:** Yani Huang, Lixin Tian, Qinghua Yang, Miaomiao Zhang, Guiyang Liu, Shaopeng Yu, Baili Feng

**Affiliations:** College of Agronomy, Northwest A & F University, Yangling, Xianyang 712100, China

**Keywords:** nitrogen fertilizer, sorghum starch, topdressing, jointing period

## Abstract

Nitrogen is a key factor affecting sorghum growth and grain quality. This experiment was designed to investigate the physicochemical properties of sorghum starch in four sorghum varieties (Liaoza 10, Liaoza 19, Jinza 31, and Jinza 34) under four nitrogen levels: 0 kg/ha urea (N1), 300 kg/ha urea as base fertilizer (N2), 300 kg/ha urea as topdressing at the jointing stage (N3), and 450 kg/ha urea as topdressing at the jointing stage (N4). The results showed that grain size and amylose content increased with increasing nitrogen fertilizer level, peaking at N3. The peak viscosity, final viscosity, gelatinization temperature, initial temperature, final temperature, and enthalpy value increased with the nitrogenous fertilizer level, peaking at N3. The application of nitrogen fertilizer at the jointing period significantly increased the above indicators. However, excess nitrogen at the jointing period (N4) can significantly reduce the above indicators, thus changing the physicochemical properties and structure of sorghum starch. Overall, nitrogen significantly affects the structure and physicochemical properties of sorghum starch.

## 1. Introduction

Sorghum is the world’s fifth most important cereal [1]. It possesses numerous desirable characteristics of high yielding, high stress, drought, waterlogging, salinity, barren, high temperature, and cold resistance, and it has a wide range of uses. Sorghum is an important drought-resistant grain crop with numerous quality varieties, grows in numerous ecological zones, and has wide utilization, including planting, food, brewing, and animal feeds [2]. In recent years, the cultivation of high-yielding and high-quality varieties and the deeper understanding of the nutritional and health value of sorghum have increased the growing of sorghum and its application. Starch is the main nutrient component in sorghum grains, and its starch content ranges from 65.3% to 81.0% but averages 79.5% [3]. However, the starch content of domestic sorghum has not received special attention, but its utilization value is far lower than that of corn, rice, and potatoes. This is mainly due to the large differences in physical and chemical properties and processing quality of sorghum starch [4]. Therefore, an in-depth understanding of the structure and properties of sorghum starch has far-reaching significance for further expansion and application of domestic sorghum starch.

Nitrogen fertilizer is the most important fertilizer for crop growth and development. Nitrogen fertilizer deficiency is the primary factor limiting crop growth, yield, and quality [5]. Numerous studies on the physicochemical properties of rice, wheat, and corn starch exist. However, only a few studies have reported the effect of nitrogen fertilizer on the physicochemical properties of sorghum starch. Nitrogen affects the crystal structure and physicochemical properties of crop starch [6]. Studies have shown that nitrogen fertilizer affects the content and quality of grain starch. Nitrogen is an important nutrient that increases the grain yield and protein content, but excess nitrogen reduces crop quality [7]. Fu et al. showed that nitrogen fertilizer application at the jointing and grain filling stages promoted the accumulation of protein and starch in grains and increased grain yield, while excessive nitrogen at this stage caused an opposite effect [8]. Alcantara et al. showed that late nitrogen fertilizer application improves rice quality by 30%–60% [9]. Nitrogen fertilizer can generally improve milling and nutrition. Gao et al. showed that nitrogen fertilizer is an important factor affecting the growth and quality of tartary buckwheat grain, in which the starch content increases with nitrogen content, peaking at 180 kgNha^−1^ [10]. However, the viscosity decreased significantly with nitrogen content, whereas the onset, peak, and conclusion first increased, and then the paste temperature and gelatinization enthalpy increased. Nitrogen fertilizer and age significantly affect the synthesis, accumulation, and physicochemical properties of buckwheat starch. Zhou et al. showed that nitrogen fertilizer affects the structure of rice starch, altering its functional properties and, thus, affecting the quality of two rice varieties [11]. Nitrogen fertilizer affects the grain quality in a quantity-dependent manner. Low to moderate nitrogen fertilizer levels improve the grain quality and size. Excess nitrogen reduces grain quality and changes rice starch’s structure and physicochemical properties. Overall, nitrogen levels significantly affect rice starch’s structure and physicochemical properties, and optimal and proper fertilization could improve rice grain quality. Zhu et al. showed that very high nitrogen decreases the size of starch granules and the amylose content [12]. Given the low order degree, nitrogen levels significantly affect the structure and physicochemical properties of rice starch.

Accordingly, in this study, we selected four Chinese sorghum varieties planted under four different nitrogen fertilizer levels and then compared their chemical compositions, physicochemical, thermal, and pasting properties. This study will provide useful information for the application of sorghum starch in food.

## 2. Materials and Methods

### 2.1. The Study Setting

The field experiment was carried out from May 2021 to October 2021 on a Small Miscellaneous Grain Test Base (38°22′ N, 109°44′ E, 1229 m above sea level) at the Academy of Agricultural Sciences, Yulin City, Shaanxi Province. This region has a temperate and semi-arid monsoon climate. The meteorological data during the sorghum growing season are shown in Figure 1, with an annual average temperature of 10.4 °C and annual average precipitation of 400 mm. Precipitation mainly occurs in August and September. The region has sandy loam soil. The previous crops were uniformly fertilized after harvesting. Base fertilizer: 2 cubic meters of cow dung; Seed fertilizer: 10 kg of diammonium hydrogen phosphate and 10 kg of potassium sulfate applied simultaneously. Urea (containing 46%) top dressing was applied at the jointing stage.

### 2.2. Plant Materials and Experimental Design

The experiments were performed using four sorghum varieties used in the brewing industry, grown in the spring-sown late-maturing areas. They included Liaoza 10, Liaoza 19, Jinza 31, and Jinza 34. The four different nitrogen fertilizer treatments were as follows: The nitrogen levels included four nitrogen treatments: 0 kg/ha urea (N1), 300 kg/ha urea as base fertilizer (N2), 300 kg/ha urea as topdressing at the jointing stage (N3), and 450 kg/ha urea as topdressing at the jointing stage (N4). All treatments applied the same amount of potassium fertilizer and phosphorus fertilizer at the same time, and the variable is the amount and period of nitrogen fertilizer application. The seeds were sown on 1 May and harvested on 11 October. Other practices were in conformity with local recommendations. A split-plot experimental design was adopted, with fertilization rate as the main plot factor and variety as the subplot factor. The treatments were performed in triplicate. The plots measured 5 m by 6 m. The row spacing was 60 cm, whereas the plant spacing was 15.5 cm (32 plants/row). The plot area was 6 × 5 × 0.60 = 18 square meters, the aisle width was 1.2 m, and the non-aisle spacing was 50 cm.

### 2.3. Determination of Key Parameter Levels

#### 2.3.1. Starch Extraction

Sorghum grain starch was extracted as previously described by Gao et al. [13]. Briefly, 500 g of sorghum grains was ground using a high-speed universal pulverizer and sieved using an 80-mesh sieve. Then, 80% ethanol was added to 200 g of sorghum flour at a ratio of 1:20 (g: mL). The flavonoids in the sorghum flour were extracted at 50 °C through ultrasonic treatment at 500 W for 30 min. Thereafter, 0.2% NaOH solution was added to the collected supernatant at a ratio of 1:10 (g: mL) at room temperature. After stirring for 15 min, the mixture was incubated at 25 °C for 22 h and filtered through a 200-mesh sieve to remove coarse fibers and other impurities to obtain coarse starch pulp. The filtrate was centrifuged in a low-speed desktop large-capacity centrifuged at 4000 r/min for 15 min. The supernatant was discarded, and the upper layer of red-brown material was collected. The above process was repeated three times. The remaining precipitate was washed three times with distilled water, and the supernatant was discarded. The pH was adjusted to 7.0 by adding distilled water and 0.1 moL/L of hydrochloric acid. After pH adjustment, the mixture was allowed to stand for 10–15 min. The supernatant was discarded, but the precipitate was transferred to a petri dish. The sorghum starch was obtained by drying in an oven at 40 degrees. The obtained solid was ground, filtered through a 100-mesh sieve, and stored at 4 °C until use.

#### 2.3.2. The Morphology of the Starch Granules

The morphology of starch granules was observed using an SEM-6360LV scanning electron microscope (JEOL, Tokyo, Japan). Briefly, a small amount of dry starch was evenly spread on the double-sided conductive adhesive of the stage and sprayed with gold using the SCD500 ion sputtering sprayer under vacuum and observed using a scanning electron microscope at a working voltage is 100 V, accelerating voltage of 15 kV, and ×1200 magnification [14].

#### 2.3.3. Granule Size Distribution

The size and distribution of starch granules were measured and observed using a laser diffraction particle size analyzer (2000 E) (Malvern Instruments, Malvern, UK). The amyloid was slowly added to the measuring beaker, and the sample granules were obtained after ultrasonic dispersion. The particulate size was determined using an in-built software in the commuter [12].

#### 2.3.4. Amylose Content

Amylose content was measured according to the method of Zhang W et al. [15]. First, 10 mL of 0.5 moL/L KOH solution was added to 0.1 g of the sorghum and boiled in a water bath for 10 min under continuous shaking. The mixture was transferred to a 50 mL volumetric flask and further mixed through shaking before picking and transferring two 2.5 mL sample solutions (i.e., sample and control solution) into a 50 mL beaker. After adding 30 mL of distilled water, the pH was adjusted to about 3.5 with 0.1 mol/HCl solution. The solution was transferred to a 50 mL volumetric flask before adding 0.5 mL iodine reagent (Preparation method: 2.0 g potassium iodide +0.2 g iodine, dilute to 100 mL using distilled water, and store in the dark). The iodine reagent was not added to the blank solution. The volume of the solutions was adjusted to 50 mL. After standing for 30 min, their absorbance was measured at 434 nm and 604 nm using a spectrophotometer. The standard curve equation was also obtained, which was expressed as y = 23.343x − 0.0385, where R = 0.9996 and y is the difference between the absorbances of the sample measured at the two wavelengths, and x is the amylose content in the sample.

#### 2.3.5. Solubility and Swelling

Solubility and swelling was measured according to the method of Gao et al. First, 10 mL of distilled water was added to 0.200 g of starch in a 15 mL centrifuge tube to prepare a 2.0% starch milk concentration. The pot was water bathed for 30 min with shaking once every 5 min. The tubes were then cooled to room temperature and centrifuged at 2000 r/min for 20 min with a large-capacity low-speed centrifuge. The supernatant was poured into an aluminum box of known mass and heated in the oven to 105 °C to a constant weight (W1). The precipitate was weighted while still in the centrifuge tube (W2) [6].

The solubility of the sorghum starch was calculated as W1/W0 × 100%; the swelling degree of sorghum starch was calculated using W2/(W0 − W1) × 100% formulae, where W0 is the dry mass of starch (g), W1 is the mass of the supernatant dried to a constant weight (g), and W2 is the wet weight of the sediment (g).

#### 2.3.6. Transparency

Transparency was measured according to the method of Gao L et al. [10]. First, exactly weigh 1 g of starch, and a starch solution with a mass concentration of 1.0% is prepared. A boiling water bath for 15 min makes it completely gelatinized, during which stirring prevents starch clumping. The light transmittance of the starch paste was measured using a photometer. Distilled water was used as the control (100% light transmittance).

#### 2.3.7. Short-Range Ordered Structure

The short-range ordered structure of sorghum starch was determined using a Fourier transform infrared spectrometer (Nicolet iS50). The starch sample was placed in the sample slot of the ATR accessory of the infrared spectrometer. The blank background was then collected, which was followed by the infrared spectrum in the range of 800 to 1200 cm^−1^. The scanning was performed 32 times at a speed of 4 cm^−1^. The peak heights at 1045 cm^−1^, 1022 cm^−1^ and 995 cm^−1^ were read from the deconvoluted spectrum, and the peak height ratio R1 (1045/1022) was between 1045 cm^−1^ and 1022 cm^−1^. The peak height ratio R2 (1022/995) was between 1022 cm^−1^ and 995 cm^−1^ [15].

#### 2.3.8. Pasting Properties

The starch moisture content was measured at room temperature using a near-infrared analyzer (DA7250). The required starch and moisture weight was calculated using a rapid viscosity analyzer (RVA-4500, Botong Ruihua Scientific Instrument (Beijing) Co., Ltd., Beijing, China). First, 2 g of starch was weighed; then, 25 mL distilled water was added, and the starch solution with a mass fraction of 8.0% was prepared by stirring evenly. The parameters were set as follows: heat at 50 °C for 1 min, heat to 95 °C within 3.7 min, maintain at 95 °C for 2.7 min, and then cool to 50 ℃ in 3.8 min. The temperature was maintained at 50 °C for 2 min. The mixture was stirred at 960 r/min in the first 10 s and at 160 r/min for the remainder of the process. The whole process lasted for 13 min. The paste temperature, peak viscosity, valley viscosity, breakage value, final viscosity, regeneration value, and peak time were analyzed using a ThermalCycle (Stockholm, Sweden) for Windows software [16].

#### 2.3.9. Thermal Properties

Thermal analysis was conducted measured using a differential scanning calorimeter (DSC) purchased from the TA Company (New Castle, Delaware, the United States) as described by Wang et al. [13]. First, 9.0 μL of ultrapure water was added to 3.0 mg of sorghum starch in an aluminum crucible. The sample was sealed/covered and incubated at room temperature for 2 h at 4 °C for 24 h to equilibrate (allow ultrapure water to infiltrate the starch thoroughly) and placed at room temperature for 1 h before testing. The measurement was carried out in DSC. The scanning was performed at 30–100 °C at a rate of 10 °C/min. An empty crucible was used as a reference. The characteristic parameters were measured and compared at sorghum starch gelatinization onset temperature (To), phase transition peak temperature (TP), phase transition termination temperature (TC) when gelatinization was complete, and heat absorbed during gelatinization (ΔH).

### 2.4. Statistical Analyses

Data were analyzed using the SPSS 19.0 statistical analysis (Armonk, NY, USA) software for a significant difference (LSD) test, whereas data processing and the drawing of graphs were performed using Origin 2021 (Northampton, MA, USA) and WPS office 2021 (Beijing, China). 

## 3. Results

### 3.1. Granules Morphology

The effect of different nitrogen fertilizer treatments on the morphology of sorghum starch granules is shown in Figure 2. The four sorghum varieties are Liaoza 10 (A1), Liaoza 19 (A2), Jinza 31 (A3) and Jinza 34 (A4).The appearance and morphology of starch in mature sorghum grain under different nitrogen fertilizer treatments were analyzed using SEM. Two types of sorghum starch granules were identified, a few of which were small spherical granules (red box), but the majority were large and irregular polygonal granules (yellow box), which is consistent with previous findings by Jane et al. who analyzed the starch morphology of 54 grains, including Sorghum grain [17].

### 3.2. Granule Size Distribution

The sorghum size under different nitrogen fertilizer levels is shown in Figure 3 and Table 1. Nitrogen fertilizer level had a significant effect on the size of sorghum starch. The distributions of starch particle size of the four sorghum varieties Liaoza 10, Liaoza 19, Jinza 31, and Jinza 34 under different nitrogen fertilizer levels showed a unimodal curve, and when the particle size is 22 μm, the proportion value is the highest. Generally, a high nitrogen fertilizer level increases the size of the starch granules. The highest effect was observed at N3, which is consistent with the findings of Gao et al. on tartary buckwheat [10].

### 3.3. Amylose Content

The effect of different nitrogen fertilizer treatments on the amylose content of sorghum grain is shown in Figure 4. Nitrogenous fertilizer had a significant effect on the amylose content of sorghum. The amylose content ranged from 16.89% to 23.43%. The amylose content of four sorghum varieties increased with nitrogen fertilizer content.

### 3.4. Solubility and Swelling

The effects of different nitrogen fertilizer treatments on the solubility and swelling of sorghum starch are shown in Figure 5. The solubility and swelling degree of starch reflect the interaction between starch and water molecules. We found that the solubility and swelling of starch increased rapidly with an increase in the temperature and were highest at 90 °C. The solubility and swelling degree of the same variety gradually decreased with the increase in nitrogen fertilizer level, and the minimum was at N3; that is, the interaction between starch and water molecules was the least. Increasing the application rate of nitrogen fertilizer in the jointing stage will increase the solubility and swelling degree. Nitrogen fertilizer significantly affects the solubility and swelling degree of sorghum starch.

### 3.5. Transparency

The effect of different nitrogen fertilizer content on the transparency of sorghum starch paste is shown in Figure 6. The transparency of starch paste reflects the dissolving and dispersion ability of starch in water. The greater the transparency, the greater the dispersion [18]. Nitrogenous fertilizer content significantly affected the transparency of sorghum starch paste, which ranged from 0.86% to 1.74%. The transparency of starch paste of Liaoza 10 and Liaoza 19 decreased with an increase in nitrogen fertilizer level. The effect of nitrogen fertilizer application at the jointing stage was obvious, but there was no significant difference after increasing nitrogen fertilizer application at the jointing stage. The transparency of the starch paste of Jinza 31 first decreased and then increased with the nitrogen fertilizer level. Increasing nitrogen fertilizer application at the jointing stage significantly reduced the transparency of the starch paste. The transparency of Jinza 34 starch paste first decreased and then increased with the nitrogen fertilizer level and was highest at N3. However, the nitrogen fertilizer level had no significant effect on the starch paste transparency if applied at the jointing stage. At this stage, the starch paste transparency first decreased and then increased with an increase in the nitrogenous fertilizer application. Generally, high nitrogen fertilizer application at the jointing stage reduced the transparency of sorghum starch paste.

### 3.6. Short-Range Ordered Structure

The Fourier transform infrared spectra of sorghum starch under different nitrogen fertilizer treatments are shown in Figure 7, and the peak height ratios R1 and R2 are shown in Table 2. Infrared spectroscopy reflects the short-range ordered structure of starch granules, and the near-infrared spectrum in the range of 1200–800 cm^−1^ is sensitive to the short-range ordered structure changes in starch. Therefore, we use ATR-FTIR to determine the short-range ordered structure of sorghum starch granules [19]. Starch contains numerous helical structures, some of which are linked by hydrogen bonds to form a stable crystalline structure, and the rest are dispersed in the amorphous region of the starch molecule. The short-range order structure of starch refers to the double-helix structure of starch. Studies have shown that the 1045/1022 cm^−1^ ratio (R1) is related to the short-range order degree of starch, and the 1022/995 cm^−1^ ratio (R2) can be used to quantify the ratio of shaped to ordered structures [20]. The highest and lowest and absorption peak of the sorghum starch (Liaoza 10, Liaoza 19, Jinza 31, and Jinza 34) was observed at 1045 cm^−1^ (blue circle) and 995 cm^−1^ (red circle). However, with the increase in nitrogen application level, the position and trend of absorption peak of infrared spectra did not change, which indicated that nitrogen application did not affect the range of the short-range ordered structure of sorghum starch.

### 3.7. Pasting Properties

Nitrogen fertilizers significantly affect the gelatinization properties of sorghum starch (Table 3). The peak viscosity (PV), final viscosity (FV), and pasting temperature (PT) of starch of four sorghum varieties Liaoza 10, Liaoza 19, Jinza 31, and Jinza 34 increased with the nitrogen application level. The effect of nitrogen fertilizer application at the jointing stage was obvious, and it peaked at N3 treatment (N3 > N2 > N1). However, a very high nitrogen level at the jointing stage significantly lowers the peak viscosity. Zhou et al. also found that with increasing nitrogen level, the peak viscosity, hot viscosity, breakdown, and final viscosity were first increased and then decreased, and the setback and pasting temperature first decreased and then increased [11]. The trough viscosity value (TV) of varieties Liaoza 10 and Liaoza 19 first increased and then decreased with an increase in the nitrogen application level. A high nitrogen fertilizer level at the jointing stage significantly reduced the valley value viscosity (N2 > N3 > N4). The viscosity of Jinza 31 and Jinza 34 increased with the nitrogenous fertilizer level. High nitrogenous fertilizer at the jointing stage significantly increased the trough viscosity, peaking at N3 (N3 > N2 > N1). However, the application of nitrogen fertilizer at the jointing stage had no significant effect on the valley viscosity. The breakdown value (BD) of Liaoza 10 and Jinza 31 first decreased and then increased with the nitrogen fertilizer level. Nitrogen fertilizer application at the jointing stage significantly increased damage, which was highest at which N3 (N3 > N1 > N2). However, very high nitrogen fertilizer at the jointing stage significantly reduced the damage and was higher at N3 than N4. The setback value (SB) of starch of the four sorghum varieties decreased with an increase in the nitrogen fertilizer level, but a very high nitrogenous fertilizer level at the jointing stage significantly increased the retrogradation value (N4 > N3).

### 3.8. Thermal Properties

The effects of different nitrogen fertilizer treatments on the enthalpy characteristics of sorghum starch are shown in Table 4. Nitrogenous fertilizer significantly affected the onset temperature (To), peak temperature (Tp), end temperature (Tc), and enthalpy value (ΔH) of starch. The starting temperature (To) and ending temperature (Tc) of starch of the four sorghum varieties changed with nitrogen application. The effect of nitrogenous fertilizer level on sorghum at the jointing stage was obvious, peaking at N3 treatment (N3 > N2 > N1, and N3 > N4). The peak temperature (Tp) first decreased and then increased with the nitrogenous fertilizer level. The effect of nitrogen fertilizer application at the jointing stage was obvious, and it peaked at N3 treatment in all the varieties (N3 > N2 > N1). However, a further increase in nitrogen fertilizer at this stage had no obvious effect on the above indexes.

## 4. Discussion

Under the conditions of our study, nitrogen fertilizer had no significant effect on the morphology of sorghum grain. The surface of starch granules in each treatment was uneven, with pits. However, the source of the pits needs further investigation. There are two possibilities: a lot of studies showed that a pit on the surface of starch particles usually exists in the contractile particles with higher α-amylase activity, that is to say A-type starch granules (wheat starch granules can be divided into two types: A and B, and A-type starch is mainly spherical, round and oval) increased in number, decreased in size, and exhibited pitting [21]. Another possibility is that pitting on the surface of starch particles is due to erosion by alkali steeping in the process of starch isolation [16]. Generally, a high nitrogen fertilizer level increases the size of the starch granules. The highest effect was observed at N3, which is consistent with the findings of Gao et al. on tartary buckwheat [10]. Top dressing at the jointing stage significantly improves the crop yield and quality [22]. However, the effect of nitrogen fertilizer on sorghum starch at the jointing stage is less studied. The N3 and N4 treatments in the present study mainly reflect the effect of topdressing on the physical and chemical properties of starch at the jointing stage. It was found that nitrogen fertilizer application at the jointing stage significantly affected the particle size distribution of sorghum starch, but the application of nitrogen fertilizer at the jointing stage should be appropriate [23]. On the contrary, (very high) nitrogen fertilizer reduces the size of sorghum starch granules (N3 > N4). The average particle size of sorghum grain for different varieties varied with the nitrogen fertilizer levels, which may be related to the specific requirements of sorghum varieties. Gao et al. found that the amylose content of tartary buckwheat first increased and then decreased with an increase in the nitrogen fertilizer level, which is consistent with the results of this experiment [10]. Zhou et al. found that the amylose and amylopectin contents of rice decreased significantly with an increase in nitrogen fertilizer application rate [11]. In addition, the ratio of amylose to amylopectin content decreased with an increase in the nitrogen level, indicating that high nitrogen levels decrease amylose synthesis in rice but increase amylopectin synthesis. Wang et al. found that the amylose content in wheat decreased with an increase in nitrogen fertilizer level, which is consistent with the increase in nitrogen application rate [24]. The inconsistent findings may be related to the unique genetic characteristics of different crops. Further experiments revealed that applying nitrogen fertilizer at the jointing period could significantly reduce the amylose content, and the effect was directly proportional to the nitrogen level (N3 > N2 > N1). Compared with the control (N1), the decrease was 1.29% and 1.28% at N3 and N2; increasing nitrogen fertilizer application at the jointing stage significantly reduced the amylose content in sorghum grain, and the effect was greater at N3 than N4 (N3 > N4). Zhu et al. found that both the solubility and swelling degree of rice starch increased with nitrogen content, which is consistent with our findings [25]. Chiotelli et al. found that the swelling degree and solubility of starch in small granules are higher than those of large granules at high nitrogen levels, whereas at low nitrogen levels, high amylose content inhibits starch granule swelling and maintains the completeness of these grains [26,27]. We found that the sorghum starch of each variety was almost insoluble at 60 °C, regardless of the nitrogenous fertilizer level [28]. The lowest starch solubility occurred at the N3 level. Increasing nitrogen fertilizer application at the jointing stage increased the solubility and swelling degree of the variety, which was highest at N4 > N3. The amylose content affects the solubility and swelling degree of starch. Nitrogen fertilizers can inhibit further starch expansion so that starch molecules can better maintain the granular seed structure [29,30]. In the present study, the amylose content of sorghum starch was high under N3 treatment, which was the same treatment in which the solubility and swelling degree were also lowest.

Zhang et al. found that the transparency of tartary buckwheat starch first increased and then decreased with an increase in the nitrogen fertilizer level, which is inconsistent with our findings on sorghum starch paste, and this could be attributed to the different genotypes among different crops [16]. The starch transparency affects the sensory perception of products and, subsequently, food’s acceptance. Therefore, this study provides a basis for developing and processing sorghum starch-related foods. Table 2 shows that the R1 value increased with the nitrogen fertilizer rate. Contrarily, R2 decreased with an increase in the nitrogen fertilizer application level. Zhou et al. found that in rice, the ratio of 1045/1022 cm^−1^ (R1) of starch first decreased and then increased, whereas the ratio of 1022/995 cm^−1^ (R2) first increased and then decreased, which is consistent with our findings [11]. In the present study, R1 increased significantly with the nitrogen fertilizer rate, indicating that nitrogen fertilizer may affect the ratio of amylose to amylopectin, which ultimately reflects the short-term ordered structure. With the increase in amylose content, amylopectin decreased and its proportion increased, which improved the order degree of starch samples [31]; R2 decreased with the increase in nitrogenous fertilizer application, indicating that the proportion of amorphous structure of sorghum starch gradually decreased, whereas high nitrogen fertilizer would increase the order degree of starch samples. The application of nitrogen fertilizer could enhance the degree of order and resistance to enzymatic hydrolysis of sorghum starch. PV is the maximum viscosity of gelatinized starch when heated in water, which reflects the expansion of starch granules [32]. The difference in peak viscosity is related to the water absorption capacity of starch granules during heating [13,14]. In this study, the PV of sorghum starch increased with the nitrogenous fertilizer level, indicating that nitrogen fertilizer increases the expansion rate of sorghum starch granules, causing rapid water absorption [33]. BD reflects the heat resistance of starch. The higher the BD value, the weaker the resistance to heating of starch. The BD value decreased with an increase in nitrogen fertilizer level at the jointing stage, indicating that high nitrogenous fertilizer level at the jointing stage improves the heat resistance of sorghum starch [34]. A high SB value indicates that the starch has a retrograde tendency. In this study, the SB value of sorghum starch decreased with an increase in the nitrogenous level, meaning that nitrogen improves the thermal stability of sorghum starch paste and prolongs its short time period of aging rate [35]. The enthalpy value (ΔH) increased with the nitrogenous fertilizer level. The effect of nitrogen fertilizer level on enthalpy value (ΔH) was obvious at the jointing stage, peaking at N3 treatment (N3 > N2 > N1), and a further increase in nitrogenous fertilizer at the jointing stage had no obvious effect on enthalpy value (ΔH). The enthalpy value (ΔH) reflects the energy consumption during the dissolution of starch. The larger the ΔH, the more energy is required for starch dissolution [36]. Kim et al. found that the high gelatinization temperature showed an indication of a higher crystalline structure in the cowpea starches that were evidenced by their higher crystallinity [37]. In the present study, the ΔH and gelatinization temperature of the starch in the four sorghum varieties increased with nitrogenous fertilizer levels, which is consistent with previous findings in rice starch [11,24,38]. The peak effect occurred at N3, which is consistent with RVA results, suggesting that excessive nitrogen decreases the dissolution of sorghum starch.

## 5. Conclusions

Under the conditions of our study, the study evaluated the effect of nitrogen levels on the structure and physicochemical properties of starch in four sorghum varieties. We found that nitrogen affects sorghum’s physicochemical properties. The particle size increased with the nitrogenous fertilizer level. A similar trend was observed for the amylose content. However, the proportion of amorphous structure in starch decreased with an increase in the nitrogenous fertilizer level. The peak viscosity, final viscosity, gelatinization temperature, initial temperature, final temperature, and enthalpy increased significantly with an increase in the nitrogenous fertilizer level. The application of nitrogenous fertilizer at the jointing period significantly increases the above parameters, all of which peaked at N3 (300 kg/ha urea as topdressing at jointing stage). However, excess nitrogen at the jointing period (N4: 450 kg/ha urea as topdressing at the jointing stage) can significantly reduce the above indicators, thus changing the physicochemical properties and structure of sorghum starch. Overall, nitrogen significantly affects the structure and physicochemical properties of sorghum starch.

## Figures and Tables

**Figure 1 foods-11-03690-f001:**
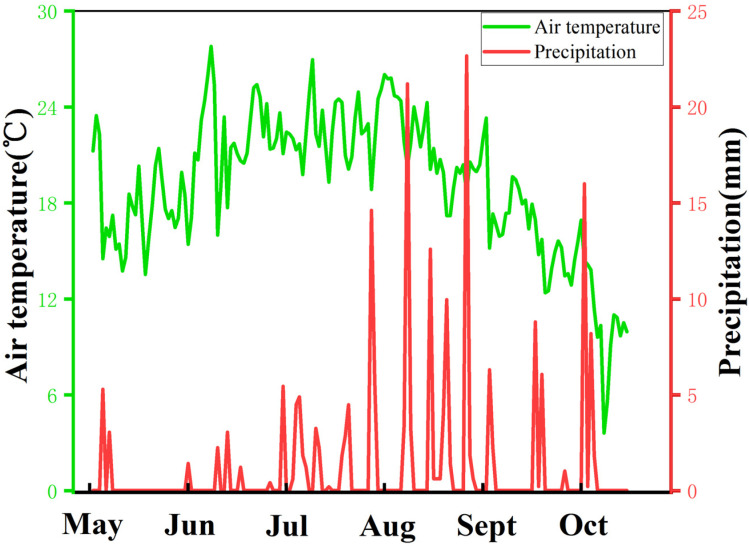
Meteorological data of sorghum growing season.

**Figure 2 foods-11-03690-f002:**
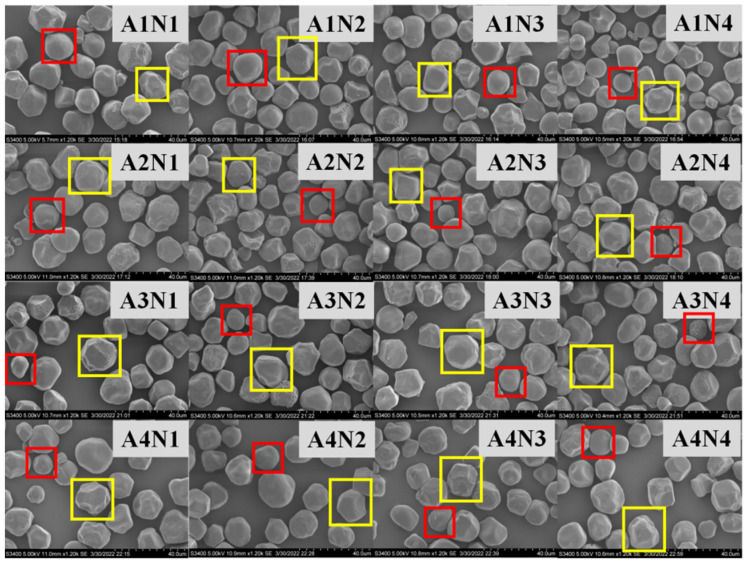
Effects of different nitrogen levels on the sorghum starch molecule morphology. (The starch molecule was observed at ×1200. Scale: 1 bar represents 40 μm.)

**Figure 3 foods-11-03690-f003:**
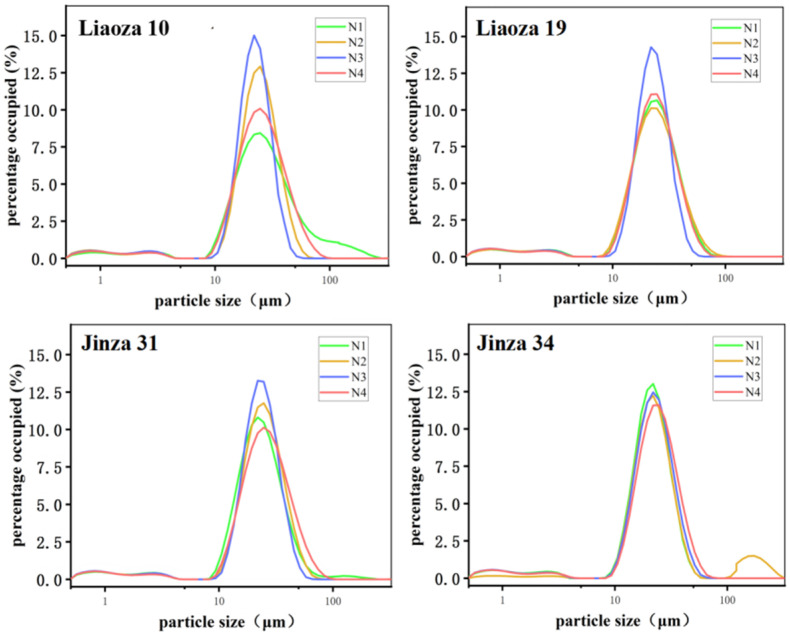
Effects of different nitrogen fertilizer levels on grain size distribution of sorghum starch.

**Figure 4 foods-11-03690-f004:**
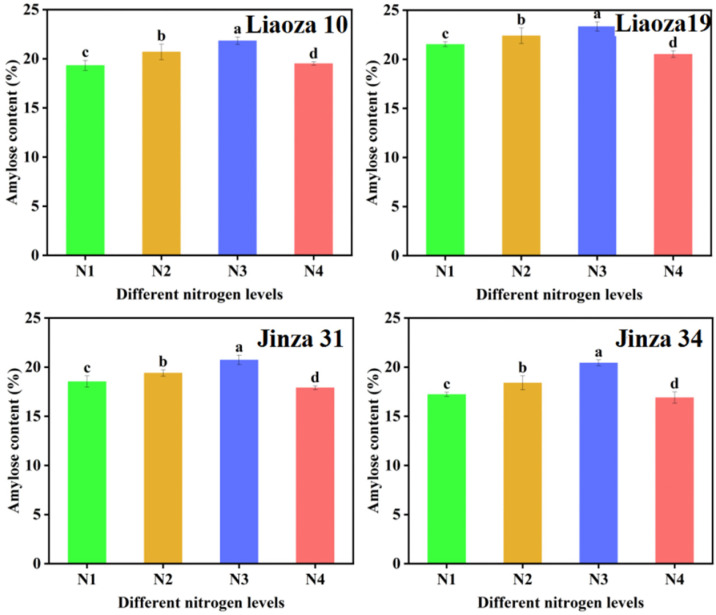
Effects of different nitrogen levels on the amylose content in sorghum grain. Note: values with letters in the figure mean a significant difference (*p* < 0.05).

**Figure 5 foods-11-03690-f005:**
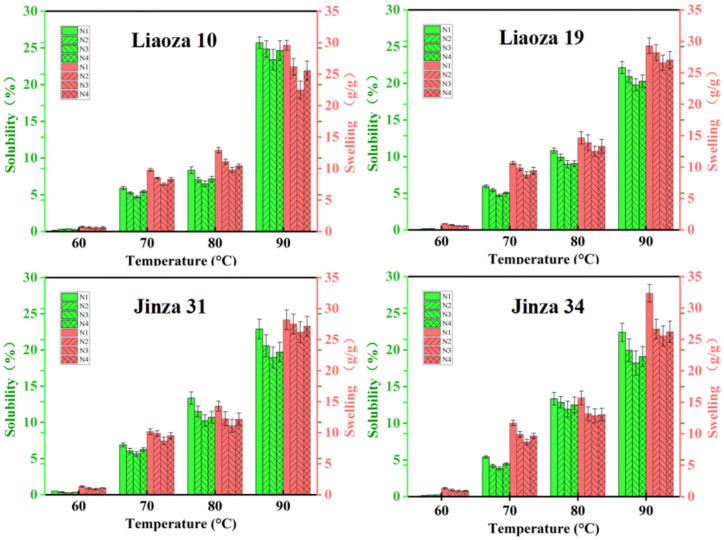
Effects of different nitrogen levels on solubility and swelling degree of sorghum starch.

**Figure 6 foods-11-03690-f006:**
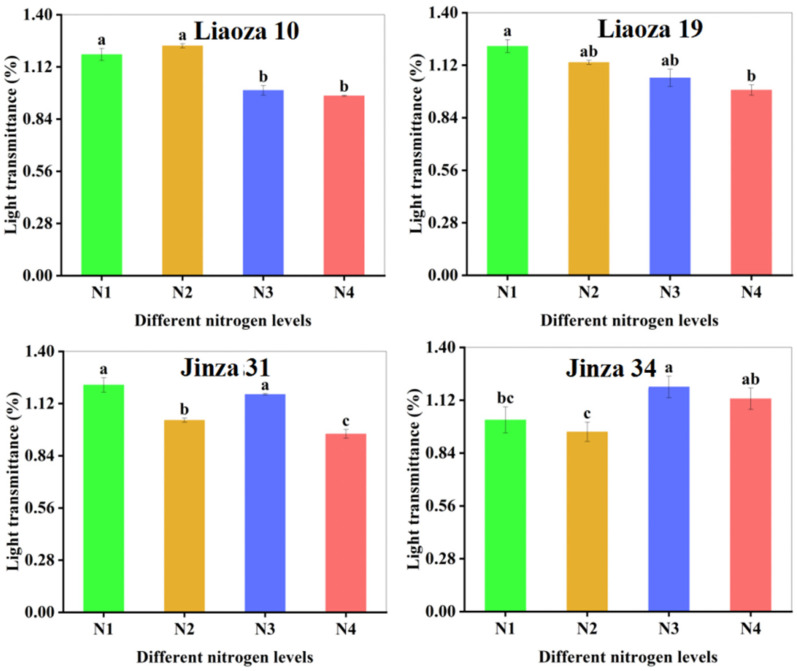
Effects of different nitrogen fertilizer level on the transparency of sorghum starch. Note: values with letters in the figure mean a significant difference (*p* < 0.05).

**Figure 7 foods-11-03690-f007:**
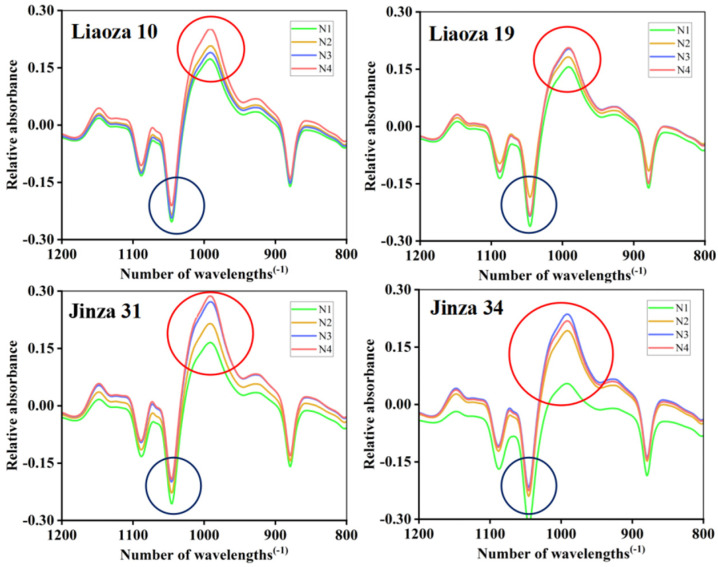
Effects of different nitrogen fertilizer level on short-range ordered structure of sorghum starch. Blue circle: 1045 cm^−1^; red circle: 995 cm^−1^.

**Table 1 foods-11-03690-t001:** Effect of nitrogenous fertilizer levels on sorghum starch morphology.

Varieties	Treatments	d (0.1)	d (0.5)	d (0.9)	D (3, 2)	D (4, 3)
Liaoza 10	N1	11.74 ± 0.19 ^c^	24.30 ± 0.64 ^a^	66.25 ± 2.83 ^a^	11.55 ± 0.25 ^a^	34.43 ± 0.97 ^a^
	N2	12.93 ± 0.17 ^a^	22.21 ± 0.35 ^b^	35.31 ± 1.50 ^c^	9.78 ± 0.11 ^c^	22.87 ± 0.57 ^c^
	N3	12.44 ± 0.08 ^b^	20.34 ± 0.18 ^c^	30.17 ± 0.38 ^d^	9.51 ± 0.07 ^c^	20.49 ± 0.21 ^d^
	N4	11.87 ± 0.20 ^c^	22.09 ± 0.62 ^b^	42.14 ± 1.56 ^b^	10.34 ± 0.17 ^b^	24.96 ± 0.77 ^b^
Liaoza 19	N1	11.56 ± 0.15 ^b^	21.89 ± 0.45 ^a^	38.43 ± 1.46 ^a^	10.05 ± 0.18 ^a^	23.36 ± 0.67 ^a^
	N2	11.17 ± 0.15 ^c^	21.67 ± 0.37 ^a^	39.63 ± 1.12 ^a^	9.87 ± 0.14 ^ab^	23.57 ± 0.52 ^a^
	N3	12.46 ± 0.12 ^a^	20.79 ± 0.31 ^b^	31.46 ± 1.29 ^b^	9.37 ± 0.11 ^c^	21.06 ± 0.49 ^b^
	N4	11.76 ± 0.26 ^b^	21.65 ± 0.17 ^a^	37.26 ± 1.53 ^a^	9.66 ± 0.09 ^bc^	22.94 ± 0.45 ^a^
Jinza 31	N1	11.16 ± 0.12 ^b^	20.81 ± 0.30 ^c^	37.48 ± 1.00 ^b^	9.72 ± 0.12 ^b^	23.87 ± 0.48 ^b^
	N2	12.38 ± 0.20 ^a^	22.23 ± 0.49 ^b^	37.01 ± 1.25 ^b^	9.74 ± 0.18 ^b^	23.26 ± 0.63 ^b^
	N3	12.48 ± 0.09 ^a^	21.32 ± 0.22 ^c^	33.23 ± 0.57 ^c^	9.51 ± 0.08 ^b^	21.80 ± 0.27 ^c^
	N4	12.14 ± 0.13 ^a^	23.37 ± 0.41 ^a^	42.82 ± 1.28 ^a^	10.12 ± 0.15 ^a^	25.43 ± 0.59 ^a^
Jinza 34	N1	11.28 ± 0.14 ^c^	19.53 ± 0.32 ^c^	30.86 ± 1.32 ^a^	8.98 ± 0.11 ^b^	20.02 ± 0.51 ^b^
	N2	12.89 ± 0.54 ^a^	21.07 ± 0.30 ^ab^	69.71 ± 4.12 ^a^	17.31 ± 6.17 ^a^	35.97 ± 3.89 ^a^
	N3	11.61 ± 0.15 ^bc^	20.39 ± 0.32 ^b^	33.02 ± 1.35 ^a^	9.19 ± 0.09 ^b^	21.11 ± 0.51 ^b^
	N4	12.06 ± 0.14 ^b^	21.64 ± 0.37 ^a^	36.24 ± 0.92 ^a^	9.62 ± 0.13 ^b^	22.72 ± 0.46 ^b^

Note: D (0.1), D (0.5), and D (0.9) represent the critical particle sizes of 10%, 50%, and 90%, respectively, of the sample population; D (3, 2) denotes the average particle surface area; D (4, 3) denotes the average particle volume. Different letters on the same column in the table indicate significant differences (*p* < 0.05).

**Table 2 foods-11-03690-t002:** Effects of different nitrogen fertilizer level on short-range ordered structure of sorghum starch.

Varieties	Treatments	R1 (1045/1022)	R2 (1022/995)
Liaoza 10	N1	3.74 ± 0.05 ^c^	0.52 ± 0.01 ^a^
	N2	2.49 ± 0.07 ^b^	0.46 ± 0.01 ^b^
	N3	2.05 ± 0.02 ^a^	0.43 ± 0.00 ^c^
	N4	1.64 ± 0.01 ^a^	0.40 ± 0.02 ^c^
Liaoza 19	N1	4.88 ± 0.13 ^c^	0.47 ± 0.03 ^b^
	N2	2.61 ± 0.03 ^b^	0.46 ± 0.04 ^b^
	N3	2.51 ± 0.04 ^a^	0.45 ± 0.02 ^a^
	N4	2.21 ± 0.02 ^a^	0.35 ± 0.04 ^a^
Jinza 31	N1	4.16 ± 0.09 ^d^	0.54 ± 0.05 ^a^
	N2	2.31 ± 0.02 ^c^	0.54 ± 0.06 ^a^
	N3	1.38 ± 0.01 ^b^	0.46 ± 0.03 ^b^
	N4	1.23 ± 0.05 ^a^	0.38 ± 0.04 ^c^
Jinza 34	N1	1.43 ± 0.08 ^c^	0.49 ± 0.05 ^a^
	N2	2.91 ± 0.07 ^b^	0.48 ± 0.03 ^b^
	N3	2.15 ± 0.03 ^a^	0.43 ± 0.01 ^c^
	N4	1.88 ± 0.06 ^a^	0.42 ± 0.01 ^c^

Note: Different letters on the same column in the table indicate significant differences (*p* < 0.05).

**Table 3 foods-11-03690-t003:** Effects of different nitrogen fertilizer level on the gelatinization of sorghum starch.

Varieties	Treatments	PV (cP)	TV (cP)	BD (cP)	FV (cP)	SB (cP)	PT (°C)
Liaoza 10	N1	4197 ± 52 ^d^	1802 ± 15 ^c^	2345 ± 14 ^d^	3242 ± 54 ^c^	1590 ± 9 ^ab^	73.35 ± 0.40 ^c^
	N2	4581 ± 62 ^b^	2086 ± 65 ^a^	2496 ± 3 ^c^	3278 ± 44 ^b^	1493 ± 21 ^b^	74.70 ± 0.45 ^b^
	N3	4796 ± 20 ^a^	1887 ± 19 ^b^	2910 ± 39 ^a^	3563 ± 23 ^a^	1494 ± 42 ^c^	76.43 ± 0.03 ^a^
	N4	4371 ± 30 ^c^	1662 ± 26 ^d^	2911 ± 4 ^b^	3275 ± 40 ^b^	1495 ± 14 ^a^	74.73 ± 0.38 ^b^
Liaoza 19	N1	5090 ± 59 ^c^	1844 ± 5 ^d^	3323 ± 34 ^a^	2884 ± 21 ^d^	1066 ± 10 ^a^	74.10 ± 0.00 ^c^
	N2	5280 ± 43 ^b^	2118 ± 11 ^a^	3324 ± 54 ^a^	2930 ± 33 ^b^	1086 ± 28 ^a^	74.73 ± 0.38 ^ab^
	N3	5388 ± 27 ^a^	1917 ± 13 ^b^	3325 ± 80 ^a^	3160 ± 21 ^a^	1046 ± 16 ^b^	75.10 ± 0.00 ^a^
	N4	5241 ± 45 ^b^	1887 ± 6 ^c^	3326 ± 71 ^a^	2921 ± 13 ^c^	1094 ± 31 ^a^	74.35 ± 0.00 ^b^
Jinza 31	N1	4263 ± 27 ^c^	2001 ± 84 ^b^	2412 ± 40 ^b^	3275 ± 49 ^c^	1174 ± 35 ^b^	73.35 ± 0.05 ^d^
	N2	4381 ± 70 ^b^	2025 ± 10 ^b^	2306 ± 10 ^c^	3387 ± 69 ^b^	1462 ± 22 ^c^	74.35 ± 0.05 ^b^
	N3	4661 ± 62 ^a^	2189 ± 5 ^a^	2472 ± 57 ^b^	3560 ± 21 ^a^	1421 ± 44 ^d^	74.85 ± 0.05 ^a^
	N4	4391 ± 38 ^b^	1798 ± 21 ^c^	2793 ± 17 ^a^	3396 ± 32 ^b^	1599 ± 16 ^a^	73.53 ± 0.03 ^c^
Jinza 34	N1	4397 ± 61 ^d^	1798 ± 9 ^c^	2599 ± 48 ^a^	3160 ± 51 ^c^	1362 ± 41 ^a^	74.30 ± 0.00 ^c^
	N2	4513 ± 23 ^b^	1811 ± 31 ^b^	2589 ± 57 ^a^	3251 ± 37 ^b^	1277 ± 34 ^b^	74.40 ± 0.00 ^b^
	N3	4619 ± 43 ^a^	1984 ± 18 ^a^	2590 ± 25 ^a^	3345 ± 11 ^a^	1361 ± 13 ^c^	75.33 ± 0.03 ^a^
	N4	4475 ± 60 ^c^	1871 ± 45 ^b^	2591 ± 16 ^a^	3165 ± 42 ^c^	1414 ± 31 ^a^	74.68 ± 0.43 ^b^

PV: peak viscosity; TV: trough viscosity; BD: breakdown; FV: final viscosity; SB: setback; PT: pasting temperature. Note: Different letters on the same column in the table indicate significant differences (*p* < 0.05).

**Table 4 foods-11-03690-t004:** Effects of different nitrogen fertilizer level on thermal enthalpy of sorghum starch.

Varieties	Treatments	To (°C)	Tp (°C)	Tc (°C)	ΔH (J/g)
Liaoza 10	N1	63.05 ± 0.21 ^c^	67.38 ± 0.01 ^c^	79.38 ± 0.13 ^ab^	9.68 ± 0.06 ^b^
	N2	63.05 ± 0.02 ^c^	67.19 ± 0.07 ^d^	78.62 ± 0.80 ^b^	9.86 ± 0.37 ^ab^
	N3	65.30 ± 0.17 ^a^	70.37 ± 0.04 ^a^	81.08 ± 0.93 ^a^	10.23 ± 0.06 ^a^
	N4	64.02 ± 0.20 ^b^	69.06 ± 0.13 ^a^	80.61 ± 1.04 ^a^	10.47 ± 0.62 ^a^
LiaoZa 19	N1	63.71 ± 0.08 ^b^	67.46 ± 0.07 ^b^	77.18 ± 0.58 ^b^	9.98 ± 0.31 ^c^
	N2	63.32 ± 0.02 ^c^	67.15 ± 0.18 ^c^	79.25 ± 0.27 ^a^	10.77 ± 0.11 ^ab^
	N3	64.40 ± 0.24 ^a^	68.42 ± 0.03 ^a^	78.53 ± 0.24 ^a^	11.50 ± 0.52 ^a^
	N4	62.79 ± 0.04 ^d^	68.24 ± 0.00 ^a^	78.43 ± 0.67 ^a^	10.32 ± 0.45 ^b^
Jinza 31	N1	62.45 ± 0.11 ^d^	67.11 ± 0.17 ^b^	75.17 ± 0.22 ^b^	8.62 ± 0.47 ^b^
	N2	63.03 ± 0.18 ^c^	66.79 ± 0.17 ^c^	75.52 ± 0.46 ^b^	8.96 ± 0.35 ^b^
	N3	63.84 ± 0.01 ^a^	68.13 ± 0.05 ^a^	76.28 ± 1.15 ^a^	9.39 ± 0.08 ^a^
	N4	63.54 ± 0.14 ^b^	67.84 ± 0.29 ^a^	76.16 ± 0.15 ^a^	9.18 ± 0.65 ^a^
Jinza 34	N1	62.72 ± 0.18 ^c^	66.98 ± 0.05 ^c^	76.63 ± 0.35 ^b^	9.94 ± 0.11 ^b^
	N2	63.51 ± 0.01 ^b^	67.59 ± 0.08 ^b^	77.17 ± 0.54 ^b^	9.94 ± 0.46 ^b^
	N3	64.38 ± 0.01 ^a^	68.41 ± 0.15 ^a^	77.55 ± 0.11 ^a^	10.04 ± 0.04 ^a^
	N4	63.40 ± 0.04 ^b^	68.31 ± 0.13 ^a^	77.00 ± 0.03 ^a^	9.94 ± 0.23 ^a^

Note: Different letters on the same column in the table indicate significant differences (*p* < 0.05).

## Data Availability

Raw sequence data are available at the National Center for Biotechnology Information Sequence Read Archive under the number SRP311252 with the run numbers SRR14000624-SRR14000704.

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
