# Peer review of "Nitrogenous Fertilizer Levels Affect the Physicochemical Properties of Sorghum Starch"

_foods, 2022, doi:10.3390/foods11223690_

Round 1

Reviewer 1 Report

1- The content of the manuscript is interesting. However, I have a concern. When doing an agronomic experiment, it needs to be repeated over two years minimum. The environmental conditions should be considered because they have a big effect on the quality of the crop. How can you explain or confirm that your results are reliable? How could you confirm that over the years the environmental conditions are not changing?

I would suggest that you add the weather data of the year 2021 and to compare it with the mean (Temperatures, rainfall) of the last X years.

Please specify in the discussion and conclusion 'under the conditions of our study'.

2- You can remove the L 71 -77 because it was repeated in the material and method.

Author Response

Thank you for the kind suggestion. Please see the attachment.

Reviewer 2 Report

The manuscript “Foods-1959902” entitled “Nitrogenous fertilizer levels affect the physicochemical properties of sorghum starch” by Huang et al. deals with an interesting subject that investigated how the nitrogenous fertilizer levels affected the physicochemical properties of sorghum starch. The experiment included four varieties (Liaoza 10, Liaoza 19, Jinza 31, and Jinza 34) and four different nitrogen fertilizer treatments as follows; 0 nitrogen fertilizer (N1); diammonium phosphate 10kg + potassium sulfate 10kg + urea 20kg, no top dressing (N2); [diammonium phosphate 10kg + potassium sulfate 10kg] + top dressing (20kg of urea at the jointing stage) (N3); [diammonium phosphate 10kg + potassium sulfate 10kg] + top dressing (30kg urea at the jointing 100 stage) (N4). The results showed that nitrogen fertilizer level had no effect on the granule morphology and of sorghum starch, but its grain size and amylose content increased with nitrogen fertilizer level, peaking at N3. The transparency first decreased and then increased with an increase in the nitrogen level. The Peak viscosity, final viscosity, gelatinization temperature, initial temperature, final temperature, and enthalpy value increased with the nitrogenous fertilizer level, peaking at N3. Moreover, application of nitrogen fertilizer at the jointing period significantly increased the above indicators. However, nitrogen fertilizer application during the jointing period is essential; otherwise, the starch quality will be very low.

For publication in “Foods”, the topic and content are appropriate. The subject of the review is interesting and topical, with high scientific and practical importance. The introduction is in accordance with the subject and correctly presented. The methodology of the study was clearly presented, and appropriate to the proposed objectives. The obtained results have been satisfactorily analyzed. The editing and linguistic quality are good. In addition, it is easy to follow by the reader, the figures and tables give good summaries, and the text editing to a thoughtful conclusion part. However, there are some points that need attention. A major revision is required for the reasons listed below:

·      The abstract is not written in a way to encourage the reader to read the whole paper. Please check and revise the treatments used in the experiment. In addition, the authors must explain the four fertilization levels of the study.

·      Keywords: Please change some keywords. The title and keywords must not contain the same words. 

·      Line 49: The correct is “Fu et al. …” instead of “Fu X Let al. ...”. The same for all of the references in your text. In general, authors should write the first author’s last name and et al.

·      Lines 72-77: It is better to refer to area as “ha” rather than as “hm2”.

·      Materials and Methods – Statistical Analysis: Authors should further explain the model of the statistical analysis used in their study. In addition, they should refer to the Post Hoc Test used for the comparison among means (e.g. Least Significant Difference (LSD) test).

·      Line 169: “2.3.8. Starch extraction” – Delete this.

·      Lines 207-209: Delete these sentences as they are instructions for the results section (MDPI template).

·      Discussion: The reference list must be enhanced since now there are only 27 references. The authors should further refer to previous studies with regard to the impacts of nitrogen fertilization on starch physicochemical properties of sorghum and other species of the Poaceae family. 

·      Please check and improve the Tables and Figures according to the journal style. In addition, increase the size of the figures, as it is very difficult to read all these graphs.

·      Finally, authors should revise the references according to the journal’s instructions for authors. 

Thank you for your consideration.

Author Response

(The authors gave the same response as above.)

Round 2

Reviewer 1 Report

All the comments have been addressed. 

Author Response

Thank you for your kind suggestion, We have addressed all the comments  according to your suggestion.

Reviewer 2 Report

The text has been corrected according to my suggestions.

Author Response

Thank you for your kind suggestion, We have corrected all the comments  according to your suggestion.